# Clinical Predictors and Prognosis of Myocardial Infarction with Non-Obstructive Coronary Arteries (MINOCA) without ST-Segment Elevation in Older Adults

**DOI:** 10.3390/jcm12031181

**Published:** 2023-02-02

**Authors:** Ana Gabaldon-Perez, Clara Bonanad, Sergio Garcia-Blas, Víctor Marcos-Garcés, Jessika Gonzalez D’Gregorio, Agustín Fernandez-Cisnal, Ernesto Valero, Gema Minana, Héctor Merenciano-González, Anna Mollar, Vicente Bodi, Julio Nunez, Juan Sanchis

**Affiliations:** 1Cardiology Department, University Hospital Clinic of Valencia, 46010 Valencia, Spain; 2INCLIVA Health Research Institute, 46010 Valencia, Spain; 3Department of Medicine, Faculty of Medicine and Odontology, University of Valencia, 46010 Valencia, Spain; 4Centro de Investigación Biomédica en Red—Cardiovascular (CIBER-CV), 28029 Madrid, Spain

**Keywords:** non-ST elevation myocardial infarction, coronary artery disease, MINOCA, older patient, prognosis

## Abstract

A non-neglectable percentage of patients with non-ST elevation myocardial infarction (NSTEMI) show non-obstructive coronary arteries (MINOCA). Specific data in older patients are scarce. We aimed to identify the clinical predictors of MINOCA in older patients admitted for NSTEMI and to explore the long-term prognosis of MINOCA. This was a single-center, observational, consecutive cohort study of older (≥70 years) patients admitted for NSTEMI between 2010 and 2014 who underwent coronary angiography. Univariate and multivariate Cox regression were performed to analyze the association of variables with MINOCA and all-cause mortality and with major adverse cardiac events (MACE), defined as a combined endpoint of all-cause mortality and nonfatal myocardial infarction and a combined endpoint of cardiovascular mortality, nonfatal myocardial infarction, and unplanned revascularization. The registry included 324 patients (mean age 78.8 ± 5.4 years), of which 71 (21.9%) were diagnosed with MINOCA. Predictors of MINOCA were female sex, left bundle branch block, pacemaker rhythm, chest pain at rest, peak troponin level, previous MI, Killip ≥2, and ST segment depression. Regarding prognosis, patients with obstructive coronary arteries (stenosis ≥50%) and the subgroup of MINOCA patients with plaques <50% had a similar prognosis; while MINOCA patients with angiographically smooth coronary arteries had a reduced risk of MACE. We conclude that the following: (1) in elderly patients admitted for NSTEMI, certain universally available clinical, electrocardiographic, and analytical variables are associated with the diagnosis of MINOCA; (2) elderly patients with MINOCA have a better prognosis than those with obstructive coronary arteries; however, only those with angiographically smooth coronary arteries have a reduced risk of all-cause mortality and MACE.

## 1. Introduction

Coronary artery disease (CAD) is one of the leading causes of mortality and morbidity and its prevalence increases with age [1,2,3]. The increase in life expectancy has led to a significant rise in the proportion of elderly patients admitted for acute coronary syndrome, with one in every three patients being over 75 years old [4]. Nevertheless, many studies have excluded elderly patients or have only included those at lower risk, leaving knowledge gaps in the diagnosis and management of CAD in this population [5,6,7,8,9,10].

Acute myocardial infarction (MI) is typically caused by acute thrombotic occlusion of a coronary artery due to atherosclerotic plaque erosion or rupture. However, approximately 5–25% of all patients presenting with acute MI have non-obstructive coronary arteries (MINOCA, defined as <50% stenosis in any epicardial coronary artery on angiography) [11,12,13,14,15,16,17,18,19,20,21]. Moreover, around two out of three patients with MINOCA present with non-ST segment elevation myocardial infarction (NSTEMI) [14,20,22]. These patients form a heterogeneous group, including several different cardiac and non-cardiac conditions [12]. The clinical characteristics of MINOCA patients are different from other patients with acute MI, although specific data in older patients are scarce [23]. It would be useful to identify predictor variables in these patients in whom it might not be necessary to opt for an invasive strategy with coronary angiography, or at least not initially [24], especially in the elderly population, which tends to have a higher incidence of complications [2,4,9].

On the other hand, discordant information has been found on the prognosis of patients presenting with MINOCA. It has been described that these patients have better [11,14,19,20,25], similar [12,17,18], or worse [12,26] outcomes than those with obstructive coronary lesions. The discrepancy in the prognosis of the patients suffering from obstructive and non-obstructive MI could reflect differences in patient profiles [27]. Analysis of variables of prognostic interest within MINOCA patients may identify a subgroup of patients who would benefit from more intensive treatment or follow-up. Moreover, MINOCA definition includes coronary arteries with <50% stenosis, but further angiographic classification may add prognostic value [24].

This study aimed to identify the clinical predictors of MINOCA in patients older than 70 years with NSTEMI and to evaluate the prognostic impact of different angiographic profiles in this population (obstructive (≥50%) coronary arteries, plaques <50%, and angiographically smooth coronary arteries).

## 2. Materials and Methods

### 2.1. Study Population

This was a single-center, observational cohort study of consecutive patients admitted for NSTEMI who underwent coronary angiography between November 2010 and February 2014. Exclusion criteria were diagnosis of cardiomyopathy, nonischemic myocardial injury (i.e., myocarditis, tako-tsubo syndrome), or noncardiac origin of the clinical picture, and previous coronary artery bypass grafting. From that cohort, we selected 324 patients aged ≥70 years for the present study (Figure 1). Data from the complete cohort were published in a previous study [24]. This project is included within the framework of the ‘‘registry of patients admitted to the cardiology ward for chest pain’’, which has been approved by the local clinical research ethics committee. Presentation and admission data were systematically collected through a specific database, including baseline clinical characteristics, cardiovascular risk factors, pharmacological treatment, electrocardiographic parameters, and CMR data. Patients were managed at the treating physician’s discretion, following clinical practice guidelines [23].

### 2.2. High-sensitivity Troponin T

Regarding troponin analysis, Elecsys high-sensitivity cardiac troponin T assay (hs-TrT, Roche Diagnostics, Switzerland) was used; 14 ng/L corresponds to the 99th percentile cutoff for a healthy reference population. A first hs-TnT determination was performed at admission, and a second one was scheduled within the first 6 h, as recommended in the current clinical guidelines at the beginning of the study period [28]. Moreover, peak hs-TnT level during admission and delta hs-TnT (estimated as the absolute or relative change between the first and second hs-TnT measurements) were also analyzed.

### 2.3. Coronary Angiography and Definition of MINOCA

All patients underwent coronary angiography during admission performed by interventional cardiologists. Obstructive coronary arteries were defined if one or more coronary stenoses ≥50% were present. MINOCA was further classified according to angiographic findings as plaques <50% and angiographically smooth coronary arteries (i.e., no angiographic evidence of coronary stenosis). MINOCA diagnosis was established according to a recent consensus document; briefly: acute MI according to the universal definition, non-obstructive coronary arteries, and exclusion of alternative causes [29].

### 2.4. CMR Study

In 52 patients, stress CMR was performed during hospital admission. A 1.5 T system (Sonata Magnetom, Siemens, Erlangen, Germany) was used. Images were acquired by a phased-array body surface coil during breath-holds and were ECG-triggered. Images were examined using customized software (Syngo, Siemens, Erlangen, Germany). Cine images were acquired in two-, three-, and four-chamber views, and in short-axis views using a steady-state free precession sequence (repetition time/echo time: 2.8/1.2 ms; flip angle: 58 degrees; matrix: 256 × 300; field of view: 320 × 270 mm; slice thickness: 7 mm). LVEF (%), LV end-diastolic volume index (ml/m2), and LV end-systolic volume index (mL/m^2^) were calculated by manual planimetry of endocardial and epicardial borders in short-axis view cine images.

Vasodilatation was induced with intravenous dipyridamole (0.84 mg/kg body weight over 6 min). After administering a gadolinium-based contrast agent, at least 3 slices in the short-axis view and 1 section in the long-axis views were acquired for hyperemia first-pass perfusion imaging using a gradient-echo sequence (inversion time: 90 ms; effective repetition time/echo time: 182 ms/1 ms; flip angle: 12°; matrix: 192 × 96; field of view: 400 × 300 mm; slice thickness: 8 mm). We analyzed the number of segments with perfusion defect on stress first-pass perfusion, i.e., a persistent delay (in at least three consecutive temporal images in comparison with other segments in the same slice) during the first pass of contrast through the myocardium after vasodilator infusion. Ischemia was disregarded in those segments exhibiting transmural late gadolinium enhancement.

Late gadolinium enhancement imaging was performed 10 min after administering the gadolinium-based contrast agent in the same locations as in the cine images using a segmented inversion recovery steady-state free precession sequence (effective repetition time/echo time: 750 ms/1.26 ms; flip angle: 45°; matrix: 256 × 184; field of view: 340 × 235 mm; slice thickness: 7 mm). Inversion time was adjusted to nullify normal myocardium. The number of segments showing transmural (>50%) necrosis was quantified using the 17-segment model.

### 2.5. Study Endpoints and Follow-Up

Two objectives were established for this study. Firstly, we aimed to identify the clinical predictors of MINOCA in older (≥70 years) patients admitted for NSTEMI. Secondly, we aimed to explore the long-term prognosis of MINOCA and its associated angiographic subtypes in this population. For this purpose, we analyzed the association of the different angiographic subtypes with the appearance of all-cause mortality and major adverse cardiac events (MACE) defined as a combined endpoint of all-cause mortality and nonfatal myocardial infarction and a combined endpoint of cardiovascular mortality, nonfatal myocardial infarction, and unplanned revascularization.

Follow-up was carried out centrally by four cardiologists using the electronic medical record of the health system in our area. The adjudication of events was carried out by consensus of the four cardiologists authorized by the local ethics committee for this purpose to confirm the event and its timing.

### 2.6. Statistical Analysis

For descriptive statistics, data were tested for normal distribution using the Kolmogorov–Smirnov test. Continuous normally distributed data were expressed as the mean ± standard deviation of the mean. Nonparametric data were expressed as the median with the interquartile range.

To identify variables associated with MINOCA diagnosis, a univariate analysis was performed using the Chi-square test or Fisher’s exact test for qualitative variables and the binary logistic regression method for quantitative ones. Then, a multivariate analysis was performed using the hierarchical multivariate logistic regression method, applying the backward stepwise methodology and including all variables that showed statistically significant or near significant associations (*p* < 0.1) in the univariate analysis. Results were expressed by the odds ratio (OR) and the 95% confidence interval (CI) (OR [CI 95%]) and statistical significance (*p*). The performance of the multivariate model was assessed by a ROC curve and by calculating the area under the curve.

To assess the independent association between MINOCA and MACE at follow-up, firstly, a univariate analysis was carried out including all study variables. Next, all the variables with a significant association (*p* < 0.05) were included in a multivariate Cox regression analysis, using the backward stepwise regression method. The results were expressed by the hazard ratio (HR) of each variable with a 95% confidence interval (95% CI) and statistical significance (*p*).

Statistical significance was achieved considering a value of *p* < 0.05. The statistical package SPSS (version 15.0, SPSS Inc., Chicago, IL, USA) and STATA (version 9.0, StataCorp, College Station, TX, USA) were used.

## 3. Results

### 3.1. MINOCA Prediction

A total of 324 patients older than 70 years (mean age 78.8 ± 5.4) admitted for NSTEMI were included, of whom 71 (21.9%) presented non-obstructive coronary arteries and were diagnosed with MINOCA. Variables associated with MINOCA in the univariate analysis are represented in Table 1 and Table 2. Briefly, MINOCA patients were more frequently female (54% vs. 36%, *p* = 0.007), while a history of prior MI, percutaneous coronary intervention (PCI), or antiplatelet treatment was less frequent in this group. With regards to the episode characteristics, MINOCA patients more frequently had a single episode of chest pain, chest pain at rest, left-bundle branch block (LBBB), and pacemaker rhythm, with fewer ST segment descents and lower hs-TnT levels.

On multivariate analysis (Table 3), several variables were independently associated with MINOCA: female sex, absence of previous MI, rest chest pain, no Killip class ≥2, no ST-segment descent, LBBB, pacemaker rhythm, and lower peak hs-TnT level. The AUC of the logistic regression model was 0.83 (95% CI: 0.78–0.88, *p* < 0.001).

### 3.2. CMR Subanalysis

In 52 patients (n = 14 with MINOCA and n = 38 with obstructive coronary arteries) CMR was performed during admission (Table 4). No differences were found between MINOCA and obstructive coronary arteries subgroups with respect to left ventricular ejection fraction and left ventricular end-systolic or end-diastolic volume index. However, patients with MINOCA showed less myocardial segments with necrosis (by late gadolinium enhancement imaging, 0 (0–0.25) segments vs. 2 (0–3.25) segments, *p* = 0.017) and less inducible ischemia (per number of segments with inducible perfusion defects, 0 (0–2) segments vs. 3 (0.75–6) segments, *p* = 0.04).

### 3.3. Prognostic Impact of Angiographic Subtypes

With a median follow-up of 5.1 years, death occurred in 43.2% (140), the combined endpoint of all-cause mortality and non-fatal MI occurred in 52.2% (n = 169), and the combined endpoint of cardiovascular mortality, non-fatal MI, and unplanned revascularization occurred in 32.4% (105) of the patients. Univariate analysis did not show statistically significant differences in all-cause mortality in patients with MINOCA compared to patients with obstructive coronary arteries (HR = 0.73 (0.52–0.95), *p* = 0.16). However, when we analyzed the combined events of all-cause mortality and non-fatal MI (HR = 0.60 (0.39–0.80), *p* = 0.013) and cardiovascular mortality, nonfatal MI, and unplanned revascularization (HR = 0.39 (0.08–0.70), *p* = 0.002), we found a significant association between MINOCA and a lower probability of MACE at follow-up (Table 5).

When analyzing angiographic subtypes independently, angiographically smooth coronary arteries were identified as a significant protective factor for all adverse events analyzed, while the plaques <50% did not show significant association for total mortality or for any of the combined endpoints (Table 5). The variables associated with the occurrence of all-cause mortality in the univariate analysis are represented in Appendix A.

Table 6 summarizes the variables that emerged as independent predictors of all-cause mortality and MACE based on the different types of events analyzed. Note that MINOCA diagnosis did not reach statistical signification. However, when the multivariate analysis was performed including MINOCA angiographic subtypes, angiographically smooth coronary arteries were independently associated with a lower occurrence of all-cause mortality and MACE (Table 6).

Figure 2 illustrates the survival curves based on the three angiographic subtypes. We can deduce that elderly patients hospitalized for NSTEMI with obstructive (≥50%) coronary arteries and the subgroup of MINOCA patients with plaques <50% depict a similar prognosis. However, MINOCA patients with angiographically smooth coronary arteries depict a reduced risk of all-cause mortality and MACE during follow-up.

## 4. Discussion

The main findings of our study constitute (a) the identification of different clinical, electrocardiographic, and analytical variables easily available during admission that allow predicting the occurrence of MINOCA in the elderly NSTEMI population; and (b) that MINOCA is associated with fewer MACE in this population but at the expense of the angiographic subtype of angiographically smooth coronary arteries, which was independently associated with a lower occurrence of MACE, behaving as a significant protective factor.

### 4.1. MINOCA in the Older Patient

The results of our study show that MINOCA is a frequent entity in older NSTEMI patients, accounting for 21.9% of our cohort. This is a somewhat high prevalence compared with other observational data, but it may reflect that more contemporary cohorts include hs-TnT assays, which are known to detect minimal myocardial damage at an expense of a lower specificity [30].

Once established that MINOCA is a frequent entity in older patients, it is interesting to describe a clinical profile associated with this diagnosis. This may help clinicians to have a more accurate diagnostic approach and to decide between an invasive or conservative strategy in older patients with NSTEMI, which is a non-completely solved issue [9]. Our objective was to determine those variables obtained with the anamnesis and the routine complementary tests used in patients with suspected acute coronary syndrome that allow us to identify those patients with very low risk of obstructive coronary arteries on angiography. Our results showed that female patients without previous MI and with a clinical presentation of single-episode rest chest pain, no signs of heart failure on admission, no ST segment changes or LBBB or pacemaker rhythm on baseline electrocardiogram, and lower hs-TnT values constitute a subgroup with a higher probability of MINOCA. Consequently, a selective invasive strategy may be preferred over a routine invasive one in this kind of patients. Nevertheless, this is just a hypothesis and further investigation is needed to support this approach.

In our cohort, only 19.7% of elderly MINOCA patients underwent CMR. However, this is very useful diagnostic tool that may help at admission in high-probability MINOCA patients to further determine the need of coronary angiography, or after MINOCA diagnosis to elucidate the specific etiology of this entity. This is reinforced by the fact that the latest clinical practice guidelines of the European Society of Cardiology for the management of NSTEMI recommend performing CMR in all MINOCA patients without an obvious underlying cause, with a recommendation class I and level of evidence B [31]. Our results showed that there are no differences in terms of volumes or ejection fraction of the left ventricle, but a statistically significant difference was observed in terms of the number of segments with late gadolinium enhancement and with inducible ischemia, being lower in the group of MINOCA patients compared with those with obstructive coronary arteries. These findings are easily interpretable, since it is expected that patients with obstructive atherosclerotic disease present greater structural involvement in the CMR.

### 4.2. Prognostic Impact of MINOCA

Considering that based on many contemporary studies MINOCA is not a benign condition and that the data available in the elderly population are scarce, we set out to identify prognostic variables in this group of patients.

The results of the multivariate analysis showed that age, personal history of diabetes, peripheral arterial disease and atrial fibrillation, previous PCI, and higher creatinine levels were independent predictors of MACE in NSTEMI elderly patients. Comorbidities and frailty may account, at least partially, for the higher MACE rate associated with age; however, a systematic evaluation of both was not performed in this study.

Univariate analysis showed a significant association between MINOCA and a lower probability of MACE at follow-up, with the exception of all-cause mortality, in which there were no statistically significant differences with respect to the obstructive coronary artery group. In contrast, in the multivariate analysis, MINOCA diagnosis did not achieve statistical significance in any of the events analyzed. However, when we analyzed the angiographic subtypes of MINOCA, we observed that the presence of angiographically smooth coronary arteries (i.e., no evidence of atherosclerosis on angiography) was an independent prognostic predictor of lower occurrence of all-cause mortality and MACE during follow-up.

Perhaps one of our most notable results is that older patients with MINOCA diagnosis and plaques <50% on angiography presented a similar prognosis to NSTEMI patients with obstructive coronary arteries. The presence of non-obstructive coronary artery disease may translate to a greater atherosclerotic burden that is related to vascular disease in other territories, so we cannot determine whether the worse prognosis is due to the progression of the coronary artery disease initially detected or to the development of clinical comorbidities derived from vascular involvement in another territory. In any case, patients with MINOCA and plaques <50% should have closer clinical follow-up and strict control of cardiovascular risk factors and comorbidities.

## 5. Study Limitations

Several limitations of this study should be acknowledged. First, there is no well-established threshold to define the geriatric population; thus, age 70 was selected as the cut-off in our cohort in accordance with the most reviewed literature. In the second place, this is a single-center study, and the results may be influenced by local peculiarities in patient management. The number of patients may limit conclusions, especially regarding subgroup analysis, which should only be considered as hypothesis generating. Angiographic classification was performed using subjective visual estimation, which may limit reproducibility. The number of patients who underwent CMR is limited. Finally, a systematic etiological study was not carried out (i.e., intracoronary imaging, microvascular function assessment, cardiac magnetic resonance, etc.) and therefore no conclusions can be drawn about the influence of the underlying pathophysiological processes on prognosis and their possible relationship with angiographic subtypes, and the lack of systematic etiological study may have led to overestimation of the MINOCA rate in the cohort.

## 6. Conclusions

In elderly patients admitted for NSTEMI, some routinely available clinical variables (female sex, absence of previous MI, rest chest pain, no Killip class ≥2, no ST-segment descent, LBBB, pacemaker rhythm, and lower peak hs-TnT level) may help predict the occurrence of MINOCA. Elderly patients with MINOCA have a better prognosis than those with obstructive coronary arteries; however, only those with angiographically smooth coronary arteries have a reduced risk of all-cause mortality and MACE, while patients with plaques <50% depict a similar prognosis to those with obstructive coronary arteries.

## Figures and Tables

**Figure 1 jcm-12-01181-f001:**
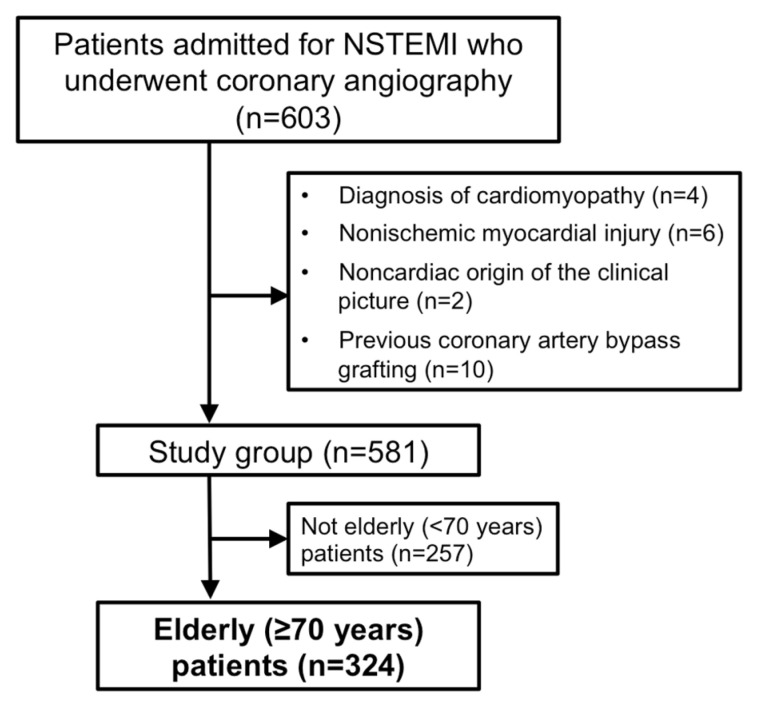
Flowchart of patients included in the study.

**Figure 2 jcm-12-01181-f002:**
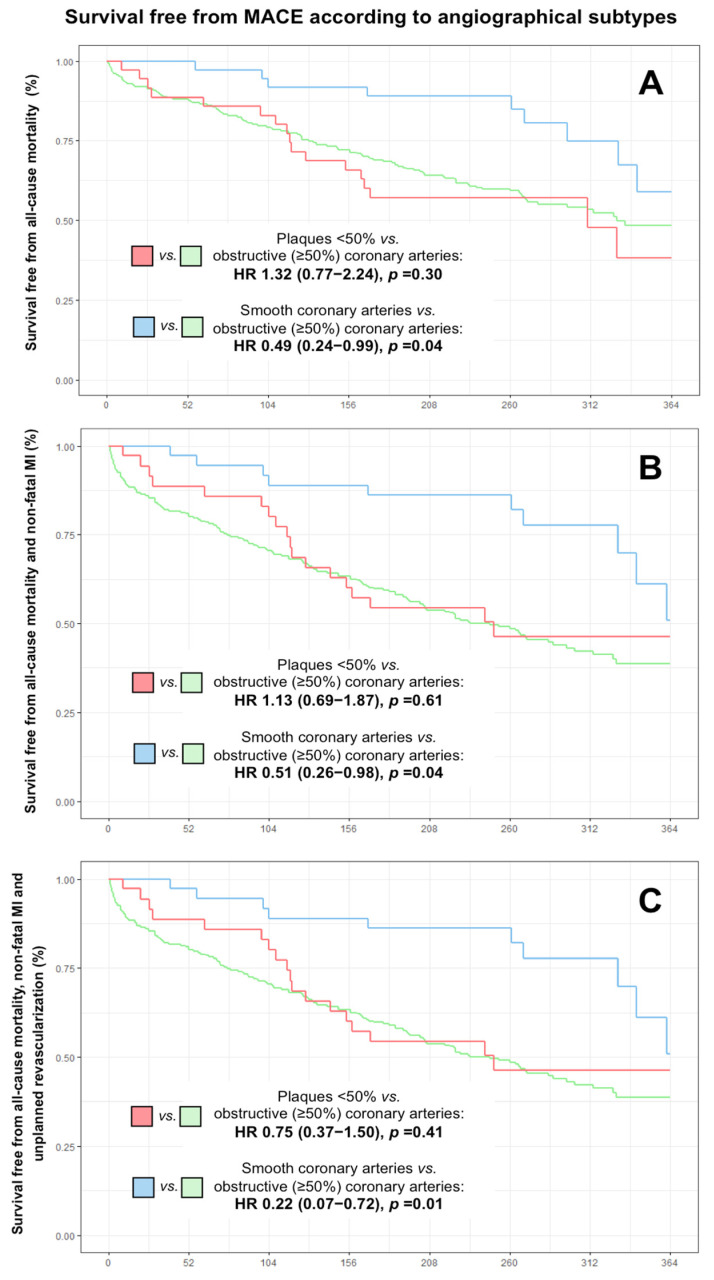
Survival free from all-cause mortality and MACE according to angiographic subtypes in elderly patients hospitalized for NSTEMI. (**A**) all-cause mortality; (**B**) all-cause mortality and non-fatal MI; (**C**) cardiovascular mortality, non-fatal MI and unplanned revascularization. Abbreviations: HR = hazard ratio; MACE = major adverse cardiac events; MI = myocardial infarction; NSTEMI = non-ST-segment elevation myocardial infarction.

**Table 1 jcm-12-01181-t001:** Clinical variables. Univariate analysis: association with MINOCA diagnosis.

	Overall Sample (N = 324)	Obstructive Coronary Arteries (N = 253)	MINOCA (N = 71)	OR (CI 95%)	*p*-Value
Age (years)	79 ± 5	79 ± 6	78 ± 5	0.98 (0.93–1.03)	0.3
Female sex (%)	128 (39)	90 (36)	38 (54)	2.09 (1.22–3.55)	0.007
Hypertension (%)	270 (83)	207 (82)	63 (89)	1.75 (0.79–3.90)	0.2
Dyslipidemia (%)	194 (60)	149 (59)	45 (63)	1.21 (0.70–2.08)	0.5
Diabetes (%)	139 (43)	114 (45)	25 (35)	0.66 (0.38–1.14)	0.1
Smoker (%)	27 (8)	24 (10)	3 (4)	0.42 (0.12–1.44)	0.2
Family history CAD (%)	4 (1)	4 (2)	0 (0)	0 (0–0)	0.9
Any CV risk factor (%)	303 (94)	237 (94)	66 (93)	0.89 (0.32–2.52)	0.8
Prior MI (%)	67 (21)	62 (25)	5 (7)	0.23 (0.09–0.61)	0.003
Prior PCI (%)	48 (15)	45 (18)	3 (4)	0.20 (0.06–0.68)	0.009
Peripheral artery disease (%)	23 (7)	19 (8)	4 (6)	0.74 (0.24–2.23)	0.6
Prior stroke (%)	31 (10)	26 (10)	5 (7)	0.66 (0.24–1.79)	0.4
Prior HF (%)	16 (5)	14 (6)	2 (3)	0.49 (0.11–2.23)	0.4
Prior antiplatelet (%)	154 (48)	129 (51)	25 (35)	0.52 (0.30–0.90)	0.02
Prior beta-blockers (%)	101 (31)	83 (33)	18 (25)	0.70 (0.38–1.26)	0.2
Prior ACE inhibitors (%)	177 (55)	141 (56)	36 (51)	0.82 (0.48–1.38)	0.5
Prior statins (%)	161 (50)	128 (51)	33 (47)	0.85 (0,50–1.44)	0.5

Abbreviations: ACE = angiotensin converting enzyme; BP = blood pressure; CAD = coronary artery disease; CI = confidence interval; CV = cardiovascular; HF = heart failure; MI = myocardial infarction; PCI = percutaneous coronary intervention.

**Table 2 jcm-12-01181-t002:** Episode characteristics. Univariate analysis: association with MINOCA diagnosis.

	Overall Sample (N = 324)	Obstructive Coronary Arteries (N = 253)	MINOCA (N = 71)	OR (CI 95%)	*p*-Value
Single episode of chest pain (%)	175 (54)	126 (50)	49 (69)	2.25 (1.28–3.93)	0.005
Rest chest pain (%)	204 (63)	148 (58)	56 (79)	2.65 (1.42–4.94)	0.002
ST descent (%)	80 (25)	76 (30)	4 (6)	0.14 (0.05–0.40)	<0.001
Negative T wave (%)	30 (9)	27 (11)	3 (4)	0.37 (0.11–1.26)	0.1
LBBB (%)	30 (9)	16 (6)	14 (20)	3.64 (1.68–7.88)	0.001
PM Rhythm (%)	11 (3)	4 (2)	7 (10)	6.81 (1.93–24)	0.003
AF (%)	40 (12)	31 (12)	9 (13)	1.04 (0.47–2.30)	0.9
Systolic BP (mmHg)	146 ± 30	145 ± 31	146 ± 27	1.001 (0.99–1.01)	0.8
Diastolic BP (mmHg)	76 ± 15	75 ± 14	78 ± 15	1.01 (0.99–1.03)	0.02
Heart rate (bpm)	80 ± 17	79 ± 17	80 ± 15	1.003 (0.99–1.02)	0.7
Killip (≥2)	44 (14)	39 (15)	5 (7)	0.42 (0.16–1.10)	0.08
Hemoglobin (g/dL)	13 ± 2	13 ± 2	13 ± 2	0.97 (0.85–1.12)	0.7
White blood cells (×10^6^/L)	8085 (6498–10,195)	8120 (6505–10,395)	7650 (6100–10,110)	1.00 (1.00–1.00)	0.6
Creatinine (mg/dL)	1.03 (0.87–1.31)	1.03 (0.88–1.31)	1 (0.80–1.37)	1.28 (0.96–1.71)	0.09
GFR (mL/min/m)	61 (46–61)	61 (46–61)	61 (42–63)	0.99 (0.98–1.01)	0.4
Peak hs-TnT level (ng/L)	78 (31–203)	97 (34–273)	39 (23–94)	0.99 (0.98–0.99)	0.007
hs-TnT absolute change	8.7 (−0.08–57)	15 (0.43–71)	2.1 (−0.82–30.38)	0.99 (0.99–1.001)	0.3

Abbreviations: AF = atrial fibrillation; BP = blood pressure; CI = confidence interval; GFR = glomerular filtration rate; LBBB = left bundle branch block; PM = pacemaker; hs-TnT = high-sensitivity troponin T.

**Table 3 jcm-12-01181-t003:** Predictors of MINOCA. Multivariate analysis.

	OR (CI 95%)	*p*-Value
Female sex	2.23 (1.18–4.22)	0.01
Previous MI	0.18 (0.06–0.53)	0.002
Rest chest pain (%)	3.33 (1.61–6.88)	0.001
ST descent (%)	0.20 (0.07–0.60)	0.004
LBBB (%)	4.54 (1.69–12.16)	0.003
PM Rhythm (%)	6.90 (1.75–27.26)	0.006
Killip ≥ 2	0.29 (0.09–0.99)	0.04
Peak troponin level	0.99 (0.98–0.99)	0.001

Abbreviations: BP = blood pressure; CI = confidence interval; LBBB = left-bundle branch block; MI = myocardial infarction; PM = pacemaker. Those variables that in the univariate analysis had shown a *p*-value ≤ 0.1 were included.

**Table 4 jcm-12-01181-t004:** CMR subanalysis in NSTEMI elderly patients with obstructive coronary arteries vs. MINOCA.

Variable	All Cohort (n = 52)	Obstructive Coronary Arteries (n = 38)	MINOCA (n = 14)	HR (CI 95%)	*p*-Value
LVEF (%)	59.79 ± 15.78	59.44 ± 16.89	60.71 ± 12.81	1.01 (0.97–1.05)	0.8
LVEDVi (mL/m^2^)	72.18 ± 28.84	74.5 ± 31.76	65.38 ± 16.99	0.99 (0.96–1.01)	0.33
LVESVi (mL/m^2^)	32.02 ± 28.21	33.21 ± 31.36	28.54 ± 16.29	0.99 (0.97–1.02)	0.61
Late gadolinium enhancement (n of segments)	1 (0–3)	2 (0–3.25)	0 (0–0.25)	0.42 (0.2–0.85)	0.017
Inducible ischemia (n of segments)	2 (0–4.5)	3 (0.75–6)	0 (0–2)	0.46 (0.22–0.97)	0.04

Abbreviations: CI = confidence interval; HR = hazard ratio; LVEDVi = left ventricular end-diastolic volume index; LVEF = left ventricular ejection fraction; LVESVi = left ventricular end-systolic volume index; MINOCA = myocardial infarction with non-obstructive coronary arteries; NSTEMI = non-ST-segment elevation myocardial infarction.

**Table 5 jcm-12-01181-t005:** Impact of angiographic subtypes on the presence of all-cause mortality and MACE during follow-up. Univariate analysis. Univariate logistic regression.

	All-Cause Mortality			
	Yes (n = 140)	No (n = 184)	HR (CI 95%)	*p*-Value
*Angiographic subtypes*				
Obstructive (≥50%) coronary arteries (%)	114 (45.1)	139 (54.9)		
MINOCA (%)	26 (36.6)	45 (63.4)	0.73 (0.48–1.13)	0.16
Plaques <50%	17 (47.2)	19 (52.8)	1.13 (0.68–1.88)	0.63
Angiographically smooth coronary arteries	9 (25.7)	26 (74.3)	0.44 (0.22–0.87)	0.019
	All-cause mortality and non-fatal MI			
	**MACE (n = 169)**	**No MACE (n = 155)**	**HR (CI 95%)**	***p*-value**
*Angiographic subtypes*				
Obstructive (≥50%) coronary arteries (%)	141 (55.7)	112 (44.2)		
MINOCA (%)	28 (39.4)	43 (60.5)	0.60 (0.39–0.90)	0.013
Plaques <50%	18 (50)	18 (50)	0.91 (0.55–1.48)	0.70
Angiographically smooth coronary arteries	10 (28.6)	25 (71.4)	0.37 (0.19–0.70)	0.002
	Cardiovascular mortality, non-fatal MI and unplanned revascularization			
	**MACE (n = 105)**	**No MACE (n = 219)**	**HR (CI 95%)**	***p*-value**
*Angiographic subtypes*				
Obstructive (≥50%) coronary arteries (%)	93 (36.8)	160 (63.2)		
MINOCA (%)	12 (16.9)	59 (83.1)	0.39 (0.21–0.72)	0.002
Plaques <50%	9 (25)	27 (75)	0.67 (0.33–1.33)	0.25
Angiographically smooth coronary arteries	3 (8.6)	32 (91.4)	0.17 (0.05–0.55)	0.003

Abbreviations: CI = confidence interval; MACE = major adverse cardiac events; MI = myocardial infarction; MINOCA = myocardial infarction with non-obstructive coronary arteries.

**Table 6 jcm-12-01181-t006:** All-cause mortality and MACE predictors. Multivariate analysis.

	HR (CI 95%)	*p*-Value
All-cause mortality		
Age (years)	1.07 (1.03–1.10)	<0.001
Prior stroke	1.94 (1.16–3.25)	0.01
Previous PCI	1.89 (1.20–3.00)	0.006
Peripheral artery disease	2.66 (1.52–4.65)	<0.001
AF	2.38 (1.53–3.70)	<0.001
GFR (mL/min/m)	0.98 (0.97–0.99)	0.02
Hemoglobin (g/dL)	0.87 (0.79–0.96)	0.009
Plaques <50% (vs. obstructive coronary arteries)	1.32 (0.77–2.24)	0.3
Angiographically smooth coronary arteries (vs. obstructive coronary arteries)	0.49 (0.24–0.99)	0.04
All-cause mortality and non-fatal MI		
Age (years)	1.08 (1.04–1.11)	<0.001
Prior stroke	1.80 (1.11–2.92)	0.01
Previous PCI	2.63 (1.75–3.96)	<0.001
Peripheral artery disease	2.15 (1.28–3.62)	0.003
AF	2.29 (1.51–3.46)	<0.001
Diabetes	1.52 (1.11–2.08)	0.008
Plaques <50% (vs. obstructive coronary arteries)	1.13 (0.69–1.87)	0.61
Angiographically smooth coronary arteries (vs. obstructive coronary arteries)	0.51 (0.26–0.98)	0.04
Cardiovascular mortality, non-fatal MI and unplanned revascularization		
Age (years)	1.03 (1.01–1.05)	0.06
Previous PCI	2.56 (1.63–4.03)	<0.001
GFR (mL/min/m)	0.98 (0.97–0.99)	0.01
Diabetes	1.91 (1.28–2.85)	0.0015
Plaques <50% (vs. obstructive coronary arteries)	0.75 (0.37–1.50)	0.41
Angiographically smooth coronary arteries (vs. obstructive coronary arteries)	0.22 (0.07–0.72)	0.01

Abbreviations: AF = atrial fibrillation; CI = confidence interval; GFR = glomerular filtration rate; MI = myocardial infarction; PCI = percutaneous coronary intervention.

## Data Availability

Data are available from the authors upon reasonable request.

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
