# Peer review of "Clinical Predictors and Prognosis of Myocardial Infarction with Non-Obstructive Coronary Arteries (MINOCA) without ST-Segment Elevation in Older Adults"

_jcm, 2023, doi:10.3390/jcm12031181_

Round 1
Reviewer 1 Report
Respected Authors,
In the beginning, I would like to acknowledge the time and energy you spent researching this topic, which I believe is indeed relevant, as there is still a lot of confusion in terms of both MINOCA patients’ treatment strategies as well as prognostic factors. However, as MINOCA is a relatively new concept, most long-term outcome studies have a number of fundamental methodological weaknesses.
As the authors themselves note, the frequency of MINOCA in the study is unusually high. This is undoubtedly due to the unadapted diagnostic algorithm used to assess MINOCA pathology (since the study was not specifically designed having MINOCA in mind), which is likely to skew the results.
- Unfortunately only “In 52 patients, stress CMR was performed during hospital admission” In 52 patients (n=14 with MINOCA and n=38 with obstructive coronary arteries) 194 CMR was performed during admission”. Therefore, a systematic etiological study was not carried out for the majority of the study’s population (intracoronary imaging, microvascular function assessment, cardiac magnetic resonance…).
Overall, the work is scientifically sound with an appropriate “general” study design, however, since the article doesn’t present significant novelties or contributions to the existing knowledge base and the MINOCA classification is questionable, I cannot reliably recommend publishing the article in its current form.
General suggestions:
The introduction sections could be improved with a more fluent transition from obstructive MI to MINOCA.
Language issues:
- Line 73: “2.1. Study poulation" seems to be miswritten. Consider replacing it.
Author Response
Reviewer 1
We would like to thank the Reviewer 1 for his/her thoughtful and constructive comments. We have re-elaborated our manuscript attending to the specific issues raised in the comments from the Reviewer. We provide a detailed list of all changes made (highlighted in yellow in the manuscript text) as well as the location of these changes in the new version of the manuscript.
Reviewer 1. Request 1
In the beginning, I would like to acknowledge the time and energy you spent researching this topic, which I believe is indeed relevant, as there is still a lot of confusion in terms of both MINOCA patients’ treatment strategies as well as prognostic factors. However, as MINOCA is a relatively new concept, most long-term outcome studies have a number of fundamental methodological weaknesses.
As the authors themselves note, the frequency of MINOCA in the study is unusually high. This is undoubtedly due to the unadapted diagnostic algorithm used to assess MINOCA pathology (since the study was not specifically designed having MINOCA in mind), which is likely to skew the results.
- Unfortunately only “In 52 patients, stress CMR was performed during hospital admission” In 52 patients (n=14 with MINOCA and n=38 with obstructive coronary arteries) 194 CMR was performed during admission”. Therefore, a systematic etiological study was not carried out for the majority of the study’s population (intracoronary imaging, microvascular function assessment, cardiac magnetic resonance…).
Overall, the work is scientifically sound with an appropriate “general” study design, however, since the article doesn’t present significant novelties or contributions to the existing knowledge base and the MINOCA classification is questionable, I cannot reliably recommend publishing the article in its current form.
Response: We are grateful to the Reviewer for his/her comments. In the “Study limitations” section, we acknowledge that a systematic etiological study was not carried out, and therefore we agree with the Reviewer’s comments regarding the study methodology. We are aware, as we clarified in the manuscript, that the lack of a systematical etiological study may have led an overestimation of the MINOCA rate in the cohort. Likewise, we understand that the number of patients who underwent CMR during admission is limited, so we have emphasized this circumstance in the “Study limitations” section (page 9, 2nd paragraph). This circumstance is probably due to the fact that at the time of inclusion of the patients the use of the CMR in this patient profile was not yet so systematized, since the recommendation to perform CMR in all MINOCA patients without an obvious underlying cause was included in the 2020 ESC clinical practice guidelines for the management of NSTEMI [1].
References:
- Collet JP, Thiele H, Barbato E, et al. 2020 ESC Guidelines for the management of acute coronary syndromes in patients presenting without persistent ST-segment elevation. Eur Heart J. 
2021;42:1289–1367.
Reviewer 1. Request 2
The introduction sections could be improved with a more fluent transition from obstructive MI to MINOCA.
Response: We would like to thank again the Reviewer's suggestions. We have considered writing the introduction focusing directly on acute myocardial infarction with non-obstructive coronary lesions, in order to provide information from the literature on this subject, both for the prediction of MINOCA and the prognosis of these patients, in order to introduce the objectives of our work.
Reviewer 1. Request 3
Line 73: “2.1. Study poulation" seems to be miswritten. Consider replacing it.
Response: We are grateful to the Reviewer for the opportunity given to correct these mistakes. We have changed the wrong word in the text of the manuscript, section “2. Material and Methods”, line 73, which now reads:
“2.1. Study population”

Reviewer 2 Report
In this article, authors shows
1)Several parameters may help to predict the occurrence of MINOCA.
2)Only MINOCA with no-atherosclerotic coronary artery have a reduced risk of MACE.
These are acceptable conclusions.
But I have several questions.
From ESC guidelene(European Heart Journal, Volume 42, Issue 14, 7 April 2021, Pages 1289–1367)
Non-obstructive coronary arteries on angiography:Defined as the absence of obstructive disease on angiography (i.e. no coronary artery stenosis ≥50%) in any major epicardial vessel.
In Table 5, 71 cases of MINOCA and 36 cases of "Non-obstructive (1-49%) coronary arteries" were listed.
The 35(71-36) cases could define as MINOCA with obstructive arteries.
What is MINOCA definition?
In Table 5, 28 cases(16.4%) of MINOCA with MACE and 43(28.1%) cases of MINOCA without MACE are listed.
16.4+28.1 = 44.5%
What is the other 55.5% cases?
I hope this is careless mistake.
Non-atherosclerotic coronary arteries has little chance of revascularization.
I am interested in whether the result is equal if MACE without revascularization is calculated.
Author Response
Reviewer 2
We would like to thank the Reviewer 2 for his/her thoughtful and constructive comments. We have re-elaborated our manuscript attending to the specific issues raised in the comments from the Reviewer. We provide a detailed list of all changes made (highlighted in yellow in the manuscript text) as well as the location of these changes in the new version of the manuscript.
Reviewer 2. Request 1
From ESC guidelene(European Heart Journal, Volume 42, Issue 14, 7 April 2021, Pages 1289–1367)
Non-obstructive coronary arteries on angiography:Defined as the absence of obstructive disease on angiography (i.e. no coronary artery stenosis ≥50%) in any major epicardial vessel.
In Table 5, 71 cases of MINOCA and 36 cases of "Non-obstructive (1-49%) coronary arteries" were listed.
The 35(71-36) cases could define as MINOCA with obstructive arteries.
What is MINOCA definition?
Response: We are grateful to the Reviewer for his/her comments. We would like to clarify that the diagnosis of MINOCA is by definition the finding of non-obstructive coronary arteries on angiography, as defined in the Materials and Methods, section “2.3. Coronary angiography and definition of MINOCA” (page 2, 7th paragraph), referring to the 2017 ESC consensus document [1]: “MINOCA diagnosis was established according to a recent consensus document; briefly: acute MI according to the universal definition, non-obstructive coronary arteries, and exclusion of alternative causes [29]”.
We decided to categorize the sample of patients without obstructive coronary lesions based on whether we observed <50% angiographic lesions or, on the contrary, they were completely smooth. However, thanks to the Reviewer's comment, we are aware that we have unfortunately not selected the proper terminology and recognize that it may lead to confusion. For this reason, we have decided to change the terminology throughout the document, dividing patients with MINOCA into the following two groups:
- Patients with plaques <50%
- Patients with smooth coronary arteries
References:
- Agewall S, Beltrame JF, Reynolds HR, et al. ESC working group position paper on 
myocardial infarction with non-obstructive coronary arteries. Eur Heart J. 
2017;38:143–153.
Reviewer 2. Request 2
In Table 5, 28 cases(16.4%) of MINOCA with MACE and 43(28.1%) cases of MINOCA without MACE are listed.
16.4+28.1 = 44.5%
What is the other 55.5% cases?
I hope this is careless mistake.
Response: We are grateful to the Reviewer for the opportunity given to correct these mistakes. We had calculated the percentage of patients with MINOCA with and without MACE with respect to the total number of patients with major events, not with respect to the total of number of patients with diagnosis of MINOCA. However, as explained in the following request, having modified the analyzed events, it is not necessary to correct any data, since the combined endpoint of all-cause mortality, non-fatal myocardial infarction and unplanned revascularization no longer belongs to the analysis. Even so, thank the reviewer for his/her comment in order not to make the same mistake again in the new data analyzed.
Reviewer 2. Request 3
Non-atherosclerotic coronary arteries has little chance of revascularization.
I am interested in whether the result is equal if MACE without revascularization is calculated.
Response: We are grateful to the Reviewer for his/her comments. Due to the comments of all the reviewers, we have considered restructuring the analysis of the prognostic impact of the angiographic subgroups, performing the analysis of all-cause mortality, the combined endpoint of all-cause mortality and non-fatal myocardial infarction, and the combined endpoint of cardiovascular mortality, nonfatal myocardial infarction and unplanned revascularization.
We have modified the following sections:
“Abstract”, page 1, 1st paragraph: “Univariate and multivariate Cox regression were performed to analyze the association of variables with MINOCA and with major adverse cardiac events (MACE), defined as all-cause mortality, a combined endpoint of all-cause mortality and nonfatal myocardial infarction and a combined endpoint of cardiovascular mortality, nonfatal myocardial infarction and unplanned revascularization”.
“Materials and Methods”, section 2.5. “Study endpoints and follow-up”, page 3, 3rd paragraph: “Secondly, to explore the long-term prognosis of MINOCA and its associated angiographic subtypes in this population. For this purpose, we analyzed the association of the different angiographic subtypes with the appearance of major adverse cardiac events (MACE) defined as all-cause mortality, a combined endpoint of all-cause mortality and nonfatal myocardial infarction and a combined endpoint of cardiovascular mortality, nonfatal myocardial infarction and unplanned revascularization”.
“Results”, section 3.3. “Prognostic impact of angiographic subtypes”, page 6-8: “With a median follow-up of 5.1 years, death occurred in 43.2% (140), the combined endpoint of all-cause mortality and non-fatal MI occurred in 52.2% (n=169), and the combined endpoint of cardiovascular mortality, non-fatal MI and unplanned revascularization occurred in 32.4% (105) of the patients. Univariate analysis did not show statistically significant differences in all-cause mortality in patients with MINOCA compared to patients with obstructive coronary arteries (HR=0.73 [0.52-0.95], p =0.16). However, when we analyzed the combined events of all-cause mortality and non-fatal MI (HR=0.60 [0.39-0.80], p =0.013) and cardiovascular mortality, nonfatal MI and unplanned revascularization (HR=0.39 [0.08-0.70], p =0.002), we found a significant association between MINOCA and a lower probability of MACE at follow-up (Table 5).
When analyzing angiographic subtypes independently, smooth coronary arteries were identified as a significant protective factor for all cardiac adverse events analyzed, while the plaques <50% did not show significant association for total mortality or for any of the combined endpoints (Table 5).
Table 6 summarizes the variables that emerged as independent predictors of MACE based on the different types of events analyzed. Note that MINOCA diagnosis did not reach statistical signification. However, when the multivariate analysis was performed including MINOCA angiographic subtypes, smooth coronary arteries were independently associated with a lower occurrence of MACE (Table 6)”.
“Table 5. Impact of angiographic subtypes on the presence of MACE during follow-up. Univariate analysis. Univariate logistic regression”, page 6:
|
|
|
|
|||||
|
|
All-cause mortality |
||||||
|
|
MACE (n=140) |
No MACE (n=184) |
HR (IC 95%) |
p-value |
|||
|
Angiographic subtypes |
|
||||||
|
Obstructive (≥50%) coronary arteries (%) |
114 (45.1) |
139 (54.9) |
|
|
|
||
|
MINOCA (%) |
26 (36.6) |
45 (63.4) |
0.73 (0.48-1.13) |
0.16 |
|
||
|
· Plaques <50% |
17 (47.2) |
19 (52.8) |
1.13 (0.68-1.88) |
0.63 |
|
||
|
· Smooth coronary arteries |
9 (25.7) |
26 (74.3) |
0.44 (0.22-0.87) |
0.019 |
|
||
|
|
All-cause mortality and non-fatal MI |
||||||
|
|
MACE (n=169) |
No MACE (n=155) |
HR (IC 95%) |
p-value |
|||
|
Angiographic subtypes |
|
||||||
|
Obstructive (≥50%) coronary arteries (%) |
141 (55.7) |
112 (44.2) |
|
|
|
||
|
MINOCA (%) |
28 (39.4) |
43 (60.5) |
0.60 (0.39-0.90) |
0.013 |
|
||
|
· Plaques <50% |
18 (50) |
18 (50) |
0.91 (0.55-1.48) |
0.70 |
|
||
|
· Smooth coronary arteries |
10 (28.6) |
25 (71.4) |
0.37 (0.19-0.70) |
0.002 |
|
||
|
|
Cardiovascular mortality, non-fatal MI and unplanned revascularization |
||||||
|
|
MACE (n=105) |
No MACE (n=219) |
HR (IC 95%) |
p-value |
|||
|
Angiographic subtypes |
|
||||||
|
Obstructive (≥50%) coronary arteries (%) |
93 (36.8) |
160 (63.2) |
|
|
|
||
|
MINOCA (%) |
12 (16.9) |
59 (83.1) |
0.39 (0.21-0.72) |
0.002 |
|
||
|
· Plaques <50% |
9 (25) |
27 (75) |
0.67 (0.33-1.33) |
0.25 |
|
||
|
· Smooth coronary arteries |
3 (8.6) |
32 (91.4) |
0.17 (0.05-0.55) |
0.003 |
|
||
Abbreviations. MACE= major adverse cardiac events; MI= myocardial infarction; MINOCA= myocardial infarction with non-obstructive coronary arteries.
“Table 6. MACE predictors. Multivariate analysis”, page 7:
|
|
HR (IC 95%) |
p-value |
|
All-cause mortality |
||
|
Age (years) Prior stroke Previous PCI Peripheral artery disease AF GFR (mL/min/m) Hemoglobin (g/dL) Plaques <50% (vs. obstructive coronary arteries) Smooth coronary arteries (vs. obstructive coronary arteries) |
1.07 (1.03-1.10) 1.94 (1.16-3.25) 1.89 (1.20-3.00) 2.66 (1.52-4.65) 2.38 (1.53-3.70) 0.98 (0.97-0.99) 0.87 (0.79-0.96) 1.32 (0.77-2.24) 0.49 (0.24-0.99) |
<0.001 0.01 0.006 <0.001 <0.001 0.02 0.009 0.30 0.04 |
|
All-cause mortality and non-fatal MI |
|
|
|
Age (years) Prior stroke Previous PCI Peripheral artery disease AF Diabetes Plaques <50% (vs. obstructive coronary arteries) Smooth coronary arteries (vs. obstructive coronary arteries) |
1.08 (1.04-1.11) 1.80 (1.11-2.92) 2.63 (1.75-3.96) 2.15 (1.28-3.62) 2.29 (1.51-3.46) 1.52 (1.11-2.08) 1.13 (0.69-1.87) 0.51 (0.26-0.98) |
<0.001 0.01 <0.001 0.003 <0.001 0.008 0.61 0.04 |
|
Cardiovascular mortality, non-fatal MI and unplanned revascularization |
||
|
Age (years) Previous PCI GFR (mL/min/m) Diabetes Plaques <50% (vs. obstructive coronary arteries) Smooth coronary arteries (vs. obstructive coronary arteries) |
1.03 (1.01-1.05) 2.56 (1.63-4.03) 0.98 (0.97-0.99) 1.91 (1.28-2.85) 0.75 (0.37-1.50) 0.22 (0.07-0.72) |
0.06 <0.001 0.01 0.0015 0.41 0.01 |
Abbreviations. AF= atrial fibrillation; GFR= glomerular filtration rate; MI= myocardial infarction; PCI= percutaneous coronary intervention.
“Figure 2. Survival free from MACE according to angiographic subtypes in elderly patients hospitalized for NSTEMI. A: all-cause mortality; B: all-cause mortality and non-fatal MI; C: cardiovascular mortality, non-fatal MI and unplanned revascularization. Abbreviations: HR= hazard ratio; MACE= major adverse cardiac events; MI= myocardial infarction; NSTEMI= non-ST-segment elevation myocardial infarction”, page 8:
Figure 2. Survival free from MACE according to angiographic subtypes in elderly patients hospitalized for NSTEMI. A: all-cause mortality; B: all-cause mortality and non-fatal MI; C: cardiovascular mortality, non-fatal MI and unplanned revascularization. Abbreviations: HR= hazard ratio; MACE= major adverse cardiac events; MI= myocardial infarction; NSTEMI= non-ST-segment elevation myocardial infarction.
“Discussion”, section “4.2. Prognostic impact of MINOCA”, page 10, 7th paragraph: “Univariate analysis showed a significant association between MINOCA and a lower probability of MACE at follow-up, with the exception of all-cause mortality, in which there were no statistically significant differences with respect to the obstructive coronary artery group. In contrast, in the multivariate analysis, MINOCA diagnosis did not achieve statistical signification in any of the events analyzed”.
“Supplementary material”, wich now reads: “Table S1 Supplementary material. Clinical variables associated to all-cause mortality. Univariate analysis”.

Reviewer 3 Report
The present paper describes predictive and prospective markers for MINOCA. It is interesting and well-written. There are two major remarks and a couple minor.
First, the classification of MINOCA is not correct. Invasive coronary angiography cannot tell if the artery is non-atherosclerotic. The correct term should be angiographically normal artery or possible the more descriptive angiographically smooth arteries.
Second, the term MACE is not correct since it includes all-cause mortality, it should be cardiac mortality. Since the investigated cohort is elderly the most interesting end-point is mortality by itself, possibly divided into CVD or not. MACE including cardiac death, AMI and revasc could be secondary end-point.
Minor remarks include:
Show a consort diagram of the selection of patients.
Tell how myocarditis and takotsubo were excluded.
Patients with a previous PCI are not considered MINOCA.
What was the rationale to exclude STEMI-patients.
Author Response
Reviewer 3
We would like to thank the Reviewer 3 for his/her thoughtful and constructive comments. We have re-elaborated our manuscript attending to the specific issues raised in the comments from the Reviewer. We provide a detailed list of all changes made (highlighted in yellow in the manuscript text) as well as the location of these changes in the new version of the manuscript.
Reviewer 3. Request 1
First, the classification of MINOCA is not correct. Invasive coronary angiography cannot tell if the artery is non-atherosclerotic. The correct term should be angiographically normal artery or possible the more descriptive angiographically smooth arteries.
Response: We are grateful to the Reviewer for his/her comments. We decided to categorize the sample of patients without obstructive coronary lesions based on whether we observed <50% angiographic lesions or, on the contrary, they were completely smooth. However, thanks to the Reviewer's comment, we are aware that we have unfortunately not selected the proper terminology and recognize that it may lead to confusion. For this reason, we have decided to change the terminology throughout the document, dividing patients with MINOCA into the following two groups:
- Patients with plaques <50%
- Patients with smooth coronary arteries
Reviewer 3. Request 2
Second, the term MACE is not correct since it includes all-cause mortality, it should be cardiac mortality. Since the investigated cohort is elderly the most interesting end-point is mortality by itself, possibly divided into CVD or not. MACE including cardiac death, AMI and revasc could be secondary end-point.
Response: We are grateful to the Reviewer for his/her comments. Due to the comments of all the reviewers, we have considered restructuring the analysis of the prognostic impact of the angiographic subgroups, performing the analysis of all-cause mortality, the combined endpoint of all-cause mortality and non-fatal myocardial infarction, and the combined endpoint of cardiovascular mortality, nonfatal myocardial infarction and unplanned revascularization.
We have modified the following sections:
“Abstract”, page 1, 1st paragraph: “Univariate and multivariate Cox regression were performed to analyze the association of variables with MINOCA and with major adverse cardiac events (MACE), defined as all-cause mortality, a combined endpoint of all-cause mortality and nonfatal myocardial infarction and a combined endpoint of cardiovascular mortality, nonfatal myocardial infarction and unplanned revascularization”.
“Materials and Methods”, section 2.5. “Study endpoints and follow-up”, page 3, 3rd paragraph: “Secondly, to explore the long-term prognosis of MINOCA and its associated angiographic subtypes in this population. For this purpose, we analyzed the association of the different angiographic subtypes with the appearance of major adverse cardiac events (MACE) defined as all-cause mortality, a combined endpoint of all-cause mortality and nonfatal myocardial infarction and a combined endpoint of cardiovascular mortality, nonfatal myocardial infarction and unplanned revascularization”.
“Results”, section 3.3. “Prognostic impact of angiographic subtypes”, page 6-8: “With a median follow-up of 5.1 years, death occurred in 43.2% (140), the combined endpoint of all-cause mortality and non-fatal MI occurred in 52.2% (n=169), and the combined endpoint of cardiovascular mortality, non-fatal MI and unplanned revascularization occurred in 32.4% (105) of the patients. Univariate analysis did not show statistically significant differences in all-cause mortality in patients with MINOCA compared to patients with obstructive coronary arteries (HR=0.73 [0.52-0.95], p =0.16). However, when we analyzed the combined events of all-cause mortality and non-fatal MI (HR=0.60 [0.39-0.80], p =0.013) and cardiovascular mortality, nonfatal MI and unplanned revascularization (HR=0.39 [0.08-0.70], p =0.002), we found a significant association between MINOCA and a lower probability of MACE at follow-up (Table 5).
When analyzing angiographic subtypes independently, smooth coronary arteries were identified as a significant protective factor for all cardiac adverse events analyzed, while the plaques <50% did not show significant association for total mortality or for any of the combined endpoints (Table 5).
Table 6 summarizes the variables that emerged as independent predictors of MACE based on the different types of events analyzed. Note that MINOCA diagnosis did not reach statistical signification. However, when the multivariate analysis was performed including MINOCA angiographic subtypes, smooth coronary arteries were independently associated with a lower occurrence of MACE (Table 6)”.
“Table 5. Impact of angiographic subtypes on the presence of MACE during follow-up. Univariate analysis. Univariate logistic regression”, page 6:
|
|
|
|
|||||
|
|
All-cause mortality |
||||||
|
|
MACE (n=140) |
No MACE (n=184) |
HR (IC 95%) |
p-value |
|||
|
Angiographic subtypes |
|
||||||
|
Obstructive (≥50%) coronary arteries (%) |
114 (45.1) |
139 (54.9) |
|
|
|
||
|
MINOCA (%) |
26 (36.6) |
45 (63.4) |
0.73 (0.48-1.13) |
0.16 |
|
||
|
· Plaques <50% |
17 (47.2) |
19 (52.8) |
1.13 (0.68-1.88) |
0.63 |
|
||
|
· Smooth coronary arteries |
9 (25.7) |
26 (74.3) |
0.44 (0.22-0.87) |
0.019 |
|
||
|
|
All-cause mortality and non-fatal MI |
||||||
|
|
MACE (n=169) |
No MACE (n=155) |
HR (IC 95%) |
p-value |
|||
|
Angiographic subtypes |
|
||||||
|
Obstructive (≥50%) coronary arteries (%) |
141 (55.7) |
112 (44.2) |
|
|
|
||
|
MINOCA (%) |
28 (39.4) |
43 (60.5) |
0.60 (0.39-0.90) |
0.013 |
|
||
|
· Plaques <50% |
18 (50) |
18 (50) |
0.91 (0.55-1.48) |
0.70 |
|
||
|
· Smooth coronary arteries |
10 (28.6) |
25 (71.4) |
0.37 (0.19-0.70) |
0.002 |
|
||
|
|
Cardiovascular mortality, non-fatal MI and unplanned revascularization |
||||||
|
|
MACE (n=105) |
No MACE (n=219) |
HR (IC 95%) |
p-value |
|||
|
Angiographic subtypes |
|
||||||
|
Obstructive (≥50%) coronary arteries (%) |
93 (36.8) |
160 (63.2) |
|
|
|
||
|
MINOCA (%) |
12 (16.9) |
59 (83.1) |
0.39 (0.21-0.72) |
0.002 |
|
||
|
· Plaques <50% |
9 (25) |
27 (75) |
0.67 (0.33-1.33) |
0.25 |
|
||
|
· Smooth coronary arteries |
3 (8.6) |
32 (91.4) |
0.17 (0.05-0.55) |
0.003 |
|
||
Abbreviations. MACE= major adverse cardiac events; MI= myocardial infarction; MINOCA= myocardial infarction with non-obstructive coronary arteries.
“Table 6. MACE predictors. Multivariate analysis”, page 7:
|
|
HR (IC 95%) |
p-value |
|
All-cause mortality |
||
|
Age (years) Prior stroke Previous PCI Peripheral artery disease AF GFR (mL/min/m) Hemoglobin (g/dL) Plaques <50% (vs. obstructive coronary arteries) Smooth coronary arteries (vs. obstructive coronary arteries) |
1.07 (1.03-1.10) 1.94 (1.16-3.25) 1.89 (1.20-3.00) 2.66 (1.52-4.65) 2.38 (1.53-3.70) 0.98 (0.97-0.99) 0.87 (0.79-0.96) 1.32 (0.77-2.24) 0.49 (0.24-0.99) |
<0.001 0.01 0.006 <0.001 <0.001 0.02 0.009 0.30 0.04 |
|
All-cause mortality and non-fatal MI |
|
|
|
Age (years) Prior stroke Previous PCI Peripheral artery disease AF Diabetes Plaques <50% (vs. obstructive coronary arteries) Smooth coronary arteries (vs. obstructive coronary arteries) |
1.08 (1.04-1.11) 1.80 (1.11-2.92) 2.63 (1.75-3.96) 2.15 (1.28-3.62) 2.29 (1.51-3.46) 1.52 (1.11-2.08) 1.13 (0.69-1.87) 0.51 (0.26-0.98) |
<0.001 0.01 <0.001 0.003 <0.001 0.008 0.61 0.04 |
|
Cardiovascular mortality, non-fatal MI and unplanned revascularization |
||
|
Age (years) Previous PCI GFR (mL/min/m) Diabetes Plaques <50% (vs. obstructive coronary arteries) Smooth coronary arteries (vs. obstructive coronary arteries) |
1.03 (1.01-1.05) 2.56 (1.63-4.03) 0.98 (0.97-0.99) 1.91 (1.28-2.85) 0.75 (0.37-1.50) 0.22 (0.07-0.72) |
0.06 <0.001 0.01 0.0015 0.41 0.01 |
Abbreviations. AF= atrial fibrillation; GFR= glomerular filtration rate; MI= myocardial infarction; PCI= percutaneous coronary intervention.
“Figure 2. Survival free from MACE according to angiographic subtypes in elderly patients hospitalized for NSTEMI. A: all-cause mortality; B: all-cause mortality and non-fatal MI; C: cardiovascular mortality, non-fatal MI and unplanned revascularization. Abbreviations: HR= hazard ratio; MACE= major adverse cardiac events; MI= myocardial infarction; NSTEMI= non-ST-segment elevation myocardial infarction”, page 8:
“Supplementary material”, wich now reads: “Table S1 Supplementary material. Clinical variables associated to all-cause mortality. Univariate analysis”.
“Discussion”, section “4.2. Prognostic impact of MINOCA”, page 10, 7th paragraph: “Univariate analysis showed a significant association between MINOCA and a lower probability of MACE at follow-up, with the exception of all-cause mortality, in which there were no statistically significant differences with respect to the obstructive coronary artery group. In contrast, in the multivariate analysis, MINOCA diagnosis did not achieve statistical signification in any of the events analyzed”.
Reviewer 3. Request 3
Show a consort diagram of the selection of patients.
Response: We agree with the Reviewer that it is important to reflect the flow of patients included in the study. Therefore, as suggested by the Reviewer, we have added a figure (“Figure 1”) in the “Material and Methods” section that describes the flow of selected patients and the reasons for exclusion, also developed in the manuscript.
“Materials and Methods”, page 2, 5th paragraph: “This was a single-center, observational cohort study of consecutive patients admitted for NSTEMI who underwent coronary angiography between November 2010 and February 2014. Exclusion criteria were diagnosis of cardiomyopathy, nonischemic myocardial injury (i.e., myocarditis, tako-tsubo syndrome) or noncardiac origin of the clinical picture, and previous coronary artery bypass grafting. From that cohort, we selected 324 patients aged ≥70 years for the present study (Figure 1)”.
Figure 1. Flowchart of patients included in the study.
Reviewer 3. Request 4
Tell how myocarditis and takotsubo were excluded.
Response: The reason for the exclusion is that these cases of non-ischemic myocardial damage are considered entities independent in themselves with a differentiated management and prognosis. Thus recommended in the ESC and AHA consensus documents on MINOCA [1, 2]. Furthermore, it is noteworthy that both myocarditis and takotsubo syndrome frequently present with ST segment elevation and in the case of myocarditis also with different symptoms, therefore, a number of patients with these diagnoses will have been previously excluded from the study group and therefore would be including in the prognostic analysis only a subgroup of patients with these entities and not the full spectrum of them.
In these cases, the diagnosis is usually carried out mainly based on imaging techniques with the criteria accepted in each case. The cardiac magnetic resonance (CMR) is a useful tool to discriminate causes of non-ischemic myocardial damage and, as recommended by the latest ESC clinical practice guidelines for the management of NSTEMI: “it is recommended to perform CMR in all MINOCA patients without an obvious underlying cause” (IB) [3]. We performed a subanalysis of the patients who underwent CMR during admission, but it was only performed on 16% of the overall sample (n=14 with MINOCA and n=38 with obstructive coronary arteries). This circumstance is probably due to the fact that at the time of inclusion of the patients these recommendations were not so well established and the use of the CMR was not so systematized in this patient profile. We know that this is one of the limitations of our work and, thanks to the Reviewer’s comment, we have the opportunity to add it in the section of “Study limitations” on Discusion (page 9, 2nd paragraph):
“The number of patients who underwent CMR is limited”.
References:
- Agewall S, Beltrame JF, Reynolds HR, et al. ESC working group position paper on 
myocardial infarction with non-obstructive coronary arteries. Eur Heart J. 
2017;38:143–153.
- Tamis-Holland JE, Jneid H, Reynolds HR, et al. Contemporary Diagnosis and 
Management of Patients With Myocardial Infarction in the Absence of Obstructive Coronary Artery Disease: A Scientific Statement From the American Heart Associ- ation. Circulation. 2019;139:e891–e908.
- Collet JP, Thiele H, Barbato E, et al. 2020 ESC Guidelines for the management of acute coronary syndromes in patients presenting without persistent ST-segment elevation. Eur Heart J. 
2021;42:1289–1367.
Reviewer 3. Request 5
Patients with a previous PCI are not considered MINOCA.
Response: We are grateful to the reviewer for his/her comments. We concur with the Reviewer that it falls within the expected that a patient with a history of AMI has coronary disease in a new event. For this reason, we could have considered excluding these patients from the study population, but we considered that they should be included since a patient with a previous AMI may have subsequent episodes of MINOCA, due to various mechanisms may coexist in the same patient at the long of the time. We believe that excluding this group would introduce a bias within the general NSTEMI population. Based on previous literature on observational studies, none of the published MINOCA series excluded patients with a history of AMI. In a cohort of 9466 patients with MINOCA from the SWEDEHEART registry, 7.6% had a previous history of AMI and other series have observed percentages between 10 and 25% [1-5]. Likewise, in the consensus documents of the ESC and the AHA, they do not mention that patients with previous PCI should be excluded [6,7].
References:
- Andersson HB, Pedersen F, Engstrøm T, Helqvist S, Jensen MK, Jørgensen E, et al. Long-term survival and causes of death in patients with ST-elevation acute coronary syndrome without obstructive coronary artery disease. Eur Heart J. 7 de enero de 2018;39(2):102-10.
- Baron T, Hambraeus K, Sundström J, Erlinge D, Jernberg T, Lindahl B. Impact on Long-Term Mortality of Presence of Obstructive Coronary Artery Disease and Classification of Myocardial Infarction. Am J Med. 1 de abril de 2016;129(4):398- 406.
- Barr PR, Harrison W, Smyth D, Flynn C, Lee M, Kerr AJ. Myocardial Infarction Without Obstructive Coronary Artery Disease is Not a Benign Condition (ANZACS-QI 10). Heart Lung Circ. 1 de febrero de 2018;27(2):165-74
- Kang WY, Jeong MH, Ahn YK, Kim JH, Chae SC, Kim YJ, et al. Are patients with angiographically near-normal coronary arteries who present as acute myocardial infarction actually safe? Int J Cardiol. 21 de enero de 2011;146(2):207-12
- Lindahl Bertil, Baron Tomasz, Erlinge David, Hadziosmanovic Nermin, Nordenskjöld Anna, Gard Anton, et al. Medical Therapy for Secondary Prevention and Long-Term Outcome in Patients With Myocardial Infarction With Nonobstructive Coronary Artery Disease. Circulation. 18 de abril de 2017;135(16):1481-9
- Agewall S, Beltrame JF, Reynolds HR, et al. ESC working group position paper on 
myocardial infarction with non-obstructive coronary arteries. Eur Heart J. 
2017;38:143–153.
- Tamis-Holland JE, Jneid H, Reynolds HR, et al. Contemporary Diagnosis and 
Management of Patients With Myocardial Infarction in the Absence of Obstructive Coronary Artery Disease: A Scientific Statement From the American Heart Associ- ation. Circulation. 2019;139:e891–e908.
Reviewer 3. Request 6
What was the rationale to exclude STEMI-patients.
Response: We would like to thank the Reviewer for the opportunity to provide an explanation in this regard. The global spectrum of AMI includes the patients with STEMI and they may have normal coronaries on angiography and, consequently, a diagnosis of MINOCA. It has been reported that approximately one third (33%, 95% CI: 22-44%; in a pooled analysis of 10 studies, n=1998) of patients with MINOCA present as STEMI [1]. On the other hand, the prevalence of normal coronaries in patients with STEMI has been reported to be 11% in a recently published series of 4793 patients [2]. The exclusion of patients with STEMI may have increased the prevalence of MINOCA with respect to other series that include it. In most of the published series they are not excluded, although even within the mentioned meta-analysis there are examples of series with only NSTEMI patients [3]. When designing this study, we decided to exclude STEMI for the following reasons: (1) the diagnosis of STEMI is evident upon arrival of the patient and the immediate approach should be urgent reperfusion in all cases, so possible predictors of MINOCA in these patients would not change this attitude, while within STEMI, if a subgroup of very high probability of MINOCA is defined, an initial conservative management without systematic coronary angiography could be chosen, and (2) patients with STEMI have a prognosis different from NSTEMI and it is influenced in a very important way by variables, such as the time of evolution of the AMI, efficacy of reperfusion therapy or extent of necrosis, which do not apply in NSTEMI or MINOCA, for which we consider that including this group in the comparison may introduce confounding factors.
References:
- Pasupathy Sivabaskari, Air Tracy, Dreyer Rachel P., Tavella Rosanna, Beltrame John F. Systematic Review of Patients Presenting With Suspected Myocardial Infarction and Nonobstructive Coronary Arteries. Circulation. 10 de marzo de 2015;131(10):861-70.
- Andersson HB, Pedersen F, Engstrøm T, Helqvist S, Jensen MK, Jørgensen E, et al. Long-term survival and causes of death in patients with ST-elevation acute coronary syndrome without obstructive coronary artery disease. Eur Heart J. 7 de enero de 2018;39(2):102-10.
- Patel MR, Chen AY, Peterson ED, Newby LK, Pollack CV, Brindis RG, et al. Prevalence, predictors, and outcomes of patients with non–ST-segment elevation myocardial infarction and insignificant coronary artery disease: Results from the Can Rapid risk stratification of Unstable angina patients Suppress ADverse outcomes with Early implementation of the ACC/AHA Guidelines (CRUSADE) initiative. Am Heart J. 1 de octubre de 2006;152(4):641-7.

Round 2
Reviewer 1 Report
The authors have taken into account the suggested remarks, however, the instruction section could be improved.
Author Response
Reviewer 1
We would like to thank the Reviewer 1 for his/her thoughtful and constructive comments. We have re-elaborated our manuscript attending to the specific issues raised in the comments from the Reviewer. We provide a detailed list of all changes made (highlighted in yellow in the manuscript text) as well as the location of these changes in the new version of the manuscript.
Reviewer 1. Request 1
The authors have taken into account the suggested remarks, however, the instruction section could be improved.
Response: We are grateful to the Reviewer for his/her comments. We have tried to improve the introduction section as required by the Reviewer (page 2, 2nd paragraph):
“Acute myocardial infarction (MI) is typically caused by acute thrombotic occlusion of a coronary artery due to atherosclerotic plaque erosion or rupture. However, approximately 5–25% of all patients presenting with acute MI have non-obstructive coronary arteries (MINOCA, defined as <50% stenosis in any epicardial coronary artery on angiography) [11–21]. Moreover, around 2 out of 3 patients with MINOCA present with non-ST segment elevation myocardial infarction (NSTEMI) [14, 20, 22]. These patients form a heterogeneous group, including several different cardiac and non-cardiac conditions [12]. The clinical characteristics of MINOCA patients are different from other patients with acute MI, although specific data in the older patient are scarce [23]. It would be useful to identify predictor variables in these patients in whom it might not be necessary to opt for an invasive strategy with coronary angiography, or at least not initially [24], especially in the elderly population, which tends to have a higher incidence of complications [2,4,9]”.

Reviewer 3 Report
Total mortality cannot be considered as a MACE, please check this in the manuscript, for example see the Tables.
Please change to angiographically smooth coronary arteries, since I am sure that no one has palpated the arteries.
Author Response
Reviewer 3
We would like to thank the Reviewer 3 for his/her thoughtful and constructive comments. We have re-elaborated our manuscript attending to the specific issues raised in the comments from the Reviewer. We provide a detailed list of all changes made (highlighted in yellow in the manuscript text) as well as the location of these changes in the new version of the manuscript.
Reviewer 3. Request 1
Total mortality cannot be considered as a MACE, please check this in the manuscript, for example see the Tables.
Response: We are grateful to the Reviewer for the opportunity given to correct these mistakes. In this regard, we have modified several sections of the manuscript as specified below:
“Abstract”, page 1, 1st paragraph:
- “Univariate and multivariate Cox regression were performed to analyze the association of variables with MINOCA and all-cause mortality and with major adverse cardiac events (MACE), defined as a combined endpoint of all-cause mortality and nonfatal myocardial infarction and a combined endpoint of cardiovascular mortality, nonfatal myocardial infarction and unplanned revascularization”.
- “We conclude that: (1) in elderly patients admitted for NSTEMI, certain universally available clinical, electrocardiographic and analytical variables are associated with the diagnosis of MINOCA; (2) elderly patients with MINOCA have a better prognosis than those with obstructive coronary arteries; however, only those with smooth coronary arteries have a reduced risk of all-cause mortality and MACE”.
“Materials and Methods”, section 2.5. “Study endpoints and follow-up”, page 3, 5th paragraph:
- “For this purpose, we analyzed the association of the different angiographic subtypes with the appearance of all-cause mortality and major adverse cardiac events (MACE) defined as a combined endpoint of all-cause mortality and nonfatal myocardial infarction and a combined endpoint of cardiovascular mortality, nonfatal myocardial infarction and unplanned revascularization”.
“Table 5”, page 7, 3rd paragraph:
- “Table 5. Impact of angiographic subtypes on the presence of all-cause mortality and MACE during follow-up. Univariate analysis. Univariate logistic regression”.
|
|
|
|
|||||||
|
|
All-cause mortality |
|
|||||||
|
|
Yes (n=140) |
No (n=184) |
HR (IC 95%) |
p-value |
|
||||
|
Angiographic subtypes |
|
|
|||||||
|
Obstructive (≥50%) coronary arteries (%) |
114 (45.1) |
139 (54.9) |
|
|
|
||||
|
MINOCA (%) |
26 (36.6) |
45 (63.4) |
0.73 (0.48-1.13) |
0.16 |
|
||||
|
17 (47.2) |
19 (52.8) |
1.13 (0.68-1.88) |
0.63 |
|
||||
|
9 (25.7) |
26 (74.3) |
0.44 (0.22-0.87) |
0.019 |
|
||||
|
|
All-cause mortality and non-fatal MI |
|
|||||||
|
|
MACE (n=169) |
No MACE (n=155) |
HR (IC 95%) |
p-value |
|||||
|
Angiographic subtypes |
|
|
|||||||
|
Obstructive (≥50%) coronary arteries (%) |
141 (55.7) |
112 (44.2) |
|
|
|
||||
|
MINOCA (%) |
28 (39.4) |
43 (60.5) |
0.60 (0.39-0.90) |
0.013 |
|
||||
|
18 (50) |
18 (50) |
0.91 (0.55-1.48) |
0.70 |
|
||||
|
10 (28.6) |
25 (71.4) |
0.37 (0.19-0.70) |
0.002 |
|
||||
|
|
Cardiovascular mortality, non-fatal MI and unplanned revascularization |
|
|||||||
|
|
MACE (n=105) |
No MACE (n=219) |
HR (IC 95%) |
p-value |
|
||||
|
Angiographic subtypes |
|
|
|||||||
|
Obstructive (≥50%) coronary arteries (%) |
93 (36.8) |
160 (63.2) |
|
|
|
||||
|
MINOCA (%) |
12 (16.9) |
59 (83.1) |
0.39 (0.21-0.72) |
0.002 |
|
||||
|
9 (25) |
27 (75) |
0.67 (0.33-1.33) |
0.25 |
|
||||
|
3 (8.6) |
32 (91.4) |
0.17 (0.05-0.55) |
0.003 |
|
||||
“Results”, page 7, 4th paragraph:
- “Table 6 summarizes the variables that emerged as independent predictors of all-cause mortality and MACE based on the different types of events analyzed”.
“Results”, page 8, 1st paragraph:
- “However, when the multivariate analysis was performed including MINOCA angiographic subtypes, smooth coronary arteries were independently associated with a lower occurrence of all-cause mortality and MACE (Table 6)”.
“Table 6”, page 8, 2nd paragraph:
- “Table 6. All-cause mortality and MACE predictors. Multivariate analysis”.
“Results”, page 8, 3rd paragraph:
- “However, MINOCA patients with smooth coronary arteries depict a reduced risk of all-cause mortality and MACE during follow-up”.
“Figure 2”, page 9:
- “Figure 2. Survival free from all-cause mortality and MACE according to angiographic subtypes in elderly patients hospitalized for NSTEMI”.
“Discussion”, section 4.2. “Prognostic impact of MINOCA”, page 10, 7th paragraph:
- “However, if we analyze the angiographic subtypes of MINOCA, we observed that the presence of smooth coronary arteries (ie, no evidence of atherosclerosis on angiography) was an independent prognostic predictor of lower occurrence of all-cause mortality and MACE during follow-up”.
“Conclusions”, page 11, 3rd paragraph:
- “Elderly patients with MINOCA have a better prognosis than those with obstructive coronary arteries; however, only those with smooth coronary arteries have a reduced risk of all-cause mortality and MACE, while patients with plaques <50% depict a similar prognosis than those with obstructive coronary arteries”.
Reviewer 3. Request 2
Please change to angiographically smooth coronary arteries, since I am sure that no one has palpated the arteries.
Response: We would like to thank again the Reviewer's suggestions. In response to the Reviewer’s comment, we have changed the term “angiographically smooth coronary arteries” throughout the entire manuscript (highlighted in yellow in the text).
